

# The response of cucumber plants (*Cucumis sativus* L.) to the application of PCB-contaminated sewage sludge and urban sediment

Anna Wyrwicka[1], Magdalena Urbaniak[2] and Mirosław Przybylski[3]

[1] Department of Plant Physiology and Biochemistry, Faculty of Biology and Environmental Protection, University of Lodz, Lodz, Poland
[2] European Regional Centre for Ecohydrology, Polish Academy of Sciences, Lodz, Poland
[3] Department of Ecology and Vertebrate Zoology, Faculty of Biology and Environmental Protection, University of Lodz, Lodz, Poland

Corresponding author
Anna Wyrwicka,
anna.wyrwicka@biol.uni.lodz.pl

## ABSTRACT

**Background**. The increasing production of sewage sludge (SS) engenders the problem of its responsible utilization and disposal. Likewise, urban sediments (SED) are deposited at the bottom of urban reservoirs and sedimentation ponds, and these require periodical dredging and utilization. However, while the SS and SED deposits often contain nutrients such as nitrogen and phosphorus; however, they also contain a variety of hazardous compounds including heavy metals, Persistent Organic Pollutants (POPs) and microbial pollutants. Fortunately, some species of Cucurbitaceae can accumulate high levels of POPs, such as polychlorinated dibenzo-*p*-dioxins (PCDD), polychlorinated dibenzofurans (PCDF) and polychlorinated biphenyls (PCB), in their tissues.

**Methods**. SS was collected from the Lodz Municipal Wastewater Treatment Plant and SED from the Sokołówka Sequential Biofiltration System. The SS and SED samples were added to soil in flower pots at three concentrations (1.8 g, 5.4 g and 10.8 g per flower pot), and one pot was left as an unamended control (C). Soil PCB concentrations were determined before cucumber planting, and after five weeks of growth. Also, total soluble protein, total chlorophyll content, chlorophyll a/b ratio and degree of lipid peroxidation (TBARS) were examined in the leaves of the cucumber plants (*Cucumis sativus* L.) cv. Cezar after five weeks. Antioxidative response was assessed by ascorbate peroxidase (APx) and catalase (CAT) assay.

**Results**. The initial PCB concentration in soil after application of SS or SED was dependent on the applied dose. After five weeks, PCB concentration fell significantly for all samples and confirmed that the dose of SS/SED had a strong effect. Soil remediation was found to be more effective after SS application. Total soluble protein content in the cucumber leaf tissues was dependent on both the type and the dose of the applied amendments, and increased with greater SS doses in the soil. The total chlorophyll content remained unchanged, and the chlorophyll a/b ratio was slightly elevated only after the application of the highest SS and SED dose. The use of SS and SED did not significantly affect TBARS content. APx activity fell after SS or SED application; however, CAT activity tended to increase, but only in the leaves of plants grown in SS-amended soil.

**Discussion**. The cultivation of cucumber plants reduces PCB concentration in soil amended with SS or SED; however, this effect is more evident in the case of SS. SS application also induced more intensive changes in the activity of enzymes engaged in antioxidative response and oxidative stress markers in plant tissues than SED. The levels of PCB in the SS may have triggered a more severe imbalance between pro- and antioxidative reactions in plants. Cucumber plants appear to be resistant to the presence of toxic substances in SS and SED, and the addition of SS and SED not only acts as a fertilizer, but also protects against accelerated aging.

# INTRODUCTION

The 21st century set a new challenge for mankind, that of reconciling economic development with the preservation and protection of global natural resources. This goal can be achieved through adopting a sustainable development strategy, which is based on fast economic growth and improvements in quality of life, coupled with the improvement of the natural environment and the protection of its resources (*European Commission, 2017*). Maintaining a balance between the above is possible only through the rational use of natural resources and limiting the environmental burden associated with waste coming from human activity. In response to the development of urban centers and the changes in consumption habits of the population, municipal waste management has become a priority environmental issue in many countries. Waste administration also represents a significant cost to society. As the global population grows, the amount of waste, including domestic wastewater, increases with it. The development of universal sewage systems requires a greater number of wastewater treatment plants, resulting in growing amounts of the product of wastewater treatment, this being sewage sludge (SS).

Urban sediments (SED) are formed at the bottom of treatment reservoirs and sedimentation ponds located in cities. One example of such structures is the Sokołówka Sequential Biofiltration System (SSBS), whose sediments were used in the present study (*Urbaniak et al., 2012*). The SSBS was constructed in the upper section of the Sokołówka River with the aim of removing sediments, suspended solids, particulate pollutants, petroleum hydrocarbons, heavy metals and nutrients from stormwater runoff through a system of sedimentation and filtration mechanisms. In order to increase the efficiency of the system, it comprises three different zones: a hydrodynamically-intensified sedimentation zone, an intensive biogeochemical process zone and an intensive biofiltration zone.

Both SS and SED are rich organic fertilizers. They contain a number of nutrients that are essential for agricultural use, such as nitrogen and phosphorus (*Herzel et al., 2016*; *Mamedov et al., 2016*; *Tontti, Poutiainen & Heinonen-Tanski, 2017*). Organic fertilization results in an increase in the organic carbon and nitrogen stock in soils and is more effective than mineral fertilization (*Yazdani et al., 2017*; *Wyrwicka & Urbaniak, 2018*). The presence

of increased amounts of soil organic matter also improves soil structure, water holding capacity and soil nutrient availability, increases microbial biomass and broadens microbial community structure and biodiversity (*Tejada et al., 2014*; *Lloreta et al., 2016*). However, in addition to substances that bestow a beneficial effect on plants and enrich the soil, SS and SED also contain heavy metals, toxic organic and inorganic pollutants, including a number of persistent organic pollutants (POPs) characterized by high persistence, potential for bioaccumulation, biomagnifications and toxicity, e.g., polychlorinated biphenyls (PCBs), polychlorinated dibenzo-*p*-dioxin (PCDD), polychlorinated dibenzofurans (PCDFs), polycyclic aromatic hydrocarbons (PAH); in addition, inorganic compounds, silicates and aluminates can also be found, as well as pathogenic and other microbial pollutants (*Oleszczuk, 2006*; *Urbaniak et al., 2017*; *Goberna & Simón, 2018*).

In accordance with sustainable development strategy principles, SS and SED should be rationally and responsibly managed so as not to pose a threat to human health and life. Landfilling, incineration, composting and agricultural use are all used for waste control in Europe. However the Landfill Directive 99/31/EC (*CEU, 1999*) and the statutes of Poland and other member countries place very strict limits on organic matter or total organic carbon content in sludge. These limits have effectively prevented the use of landfilling as a means of disposal. Hence, the first choice for sludge disposal indicated by European Environmental Policy (Decision 2001/118/EC; Directive 2008/98/EC; *CEC, 2001*) is total sludge reuse, with agricultural use being the obvious mode. However, due to the elevated concentrations of heavy metals in SS, which are regulated at national (Journal of Laws of 2015, item 257) and EU levels (Sewage Sludge Directive, 86/ 278/EEC; *CEC, 1986*), this approach cannot be used by agriculture for food production.

A method that addresses the problem of increasing amounts of SS and SED is phytoremediation. It is a technique based on the ability of selected plant species to remove, contain or render harmless environmental contaminants (*Cunningham & Berti, 1993*; *Cunningham, Berti & Huang, 1995*; *Macek, Macková & Káš, 2000*). This ''green technology'' is a cost-effective, non-invasive, alternative or complementary technology for site restoration and partial decontamination (*Wiesmeier et al., 2015*; *Glenn, Jordan & Waugh, 2017*). Phytoremediation is a promising method of coping with the growing production of SS and urban reservoir sediments, particularly in cities, where they could be used by direct application as soil additives on city gardens and lawns. The plants used in phytoremediation should not only have a high ability to neutralize pollutants, but should also be resistant to the effects of these pollutants on plant organisms.

Although many examples have been given of plant species capable of remediation of heavy metals (*Ivanov, Bystrova & Seregin, 2003*; *Reddy et al., 2005*; *Lyubenova et al., 2009*), the processes of phytoremediation of organic substances are not so well understood and further information is still needed on this subject. The main problem associated with the potential for phytoremediation of organic compounds is related to their hydrophobicity, and hence the inability of plants to take up these compounds from the soil (log $K_{ow}$ between 5.0 to 8.3). Substances with a log $K_{ow}$ higher than 3.5 are not available for most plants, because they are strongly bound to soil particles and do not go into the soil solution from where they may be taken up by the roots (*Briggs, Bromilow & Evans, 1982*; *Hatzinger &*
*Alexander, 1995*). Many excellent publications indicate that the plants of the Cucurbitaceae, such as cucumbers, pumpkins and squashes are exceptions to this rule. These plants have a high potential to take in relatively large amounts of organic compounds and to accumulate them in their tissues. In addition, it was found that the collected compounds come largely from the substrate in which the plants were grown rather than from the air (*Hülster, Müller & Marschner, 1994*; *Engwall & Hjelm, 2000*; *Mattina, Iannucci-Berger & Dykas, 2000*; *White, 2002*; *Mattina et al., 2004*; *Zhang et al., 2009*; *Inui et al., 2008*; *Low et al., 2010*; *White, 2010*; *Matsuo et al., 2011*).

SS and SED offer great potential as fertilizers which also improve the physicochemical properties of soil due to their high organic matter content. On the other hand, the presence of harmful substances in the sediments may adversely affect plant physiology and influence total soluble protein content, total chlorophyll content as well as chlorophyll a/b ratio, among others. However, most research on the effects of SS or SED on plants focuses mainly on growth parameters such as biomass yield (*Urbaniak, Zieliński & Wyrwicka, 2017*). Only a few works investigate the influence of these sediments on physiological and biochemical changes occurring in treated plants (*Lakhdar et al., 2009*).

Environmental stress factors have been demonstrated to result in delayed secondary oxidative stress in plants (*Demidchik, 2015*). This state is characterized by the imbalance between the formation of Reactive Oxygen Species (ROS) and their removal by the cellular antioxidant systems. ROS such as superoxide anions ($O_2^{-\cdot}$), hydroxyl radicals ($\cdot OH$) and peroxyl radicals ($ROO^\cdot$), singlet oxygen ($^1O_2$) and hydrogen peroxide ($H_2O_2$) are highly reactive and can cause oxidative damage to proteins, lipids, nucleic acids and other biologically important molecules; this damage results in dysfunction and irreversible degradation, and thus reduced plant growth and development (*Gill & Tuteja, 2010*). Although ROS are naturally formed under physiological conditions in plant cells as by-products of photosynthesis and respiration, the homeostasis between their production and elimination is not maintained under oxidative stress. One of the consequences of the occurrence of oxidative stress in plant tissues is the increase in the concentration of compounds reacting with thiobarbituric acid (ThioBarbituric Acid Reactive Substances; TBARS). In plant cells, many elements of the antioxidant system interact with each other to counteract ROS accumulation. One of the components of the enzymatic antioxidant system is ascorbate peroxidase (APx), which reduces $H_2O_2$ to water with simultaneous oxidation of the substrate, ascorbic acid. Another antioxidant enzyme is catalase (CAT), which catalyzes the $H_2O_2$ dismutation reaction.

The aim of the study was to determine the effect of SS and SED contamination of soil on cucumber cultivation, with regard to their potential as soil remediation agents. The study examines total soluble protein and total chlorophyll content, and chlorophyll a/b ratio in the plants to identify the physiological status. To determine if the given conditions affect the redox equilibrium, the extent of the oxidative reaction was assessed as TBARS content, and efficiency of the antioxidative system was assessed as the activities of APx and CAT.

**Table 1  Physico-chemical properties of the untreated and soil amended with SS at the dose of 9 t ha$^{-1}$.**

| Properties | Unit | Control soil | Soil mixed with 9 t ha$^{-1}$ of SS |
|---|---|---|---|
| pH | – | 6.02 | 6.4 |
| total organic carbon | g kg$^{-1}$ | 15.18 | 22.67 |
| total nitrogen | % | 0.52 | 0.66 |
| N-NO$_3$ | mg kg$^{-1}$ d.w. | 782 | 859 |
| P | mg kg$^{-1}$ d.w. | 437 | 728 |
| K | mg kg$^{-1}$ d.w. | 2,341 | 2,809 |

## MATERIAL AND METHODS

### Soil preparation

Two sets of samples were collected from two locations in Central Poland: SS from Lodz Municipal Wastewater Treatment Plant (LM WWTP) and SED from the Sokołówka Sequential Biofiltration System (SSBS). The SED samples used in the experiment were collected from the first zone of the SSBS, intended for accelerated sedimentation of suspended matter and associated pollutants. The SS and SED samples were dried at 70 °C for 72 h then homogenized into small particles using a mortar and used as fertilizer for the soil samples for cucumber planting. The vegetable potting soil, intended for cucumber growth, used in the experiment was collected from Hollas Sp. z o.o. Pasłęk. The main soil parameters used in this experiment was presented in Table 1.

In addition to controls, in which no sludge or sediment was added (C), three treatments were used: 1.8 g, 5.4 g and 10.8 g of dried matter per flower pot. The first corresponds to a dose of 3 tonnes ha$^{-1}$, the annual dosage permitted by the Ministry of the Environment of Regulations (Journal of Laws of 2015, item 257); the second is the permitted dose of 9 tonnes ha$^{-1}$ per three years applied on one occasion; and the third, 18 tonnes ha$^{-1}$ is above the permitted level. Treatments are designated by the dose per pot.

### PCB concentrations

PCB concentrations were determined in the control and SS and SED amended soils before cucumber planting, i.e., the first day of the experiment, and after five weeks of cucumber growth, i.e., the last day of the experiment. The collected soil samples were stored in glass tubes in the dark at 4 °C for further extraction using a PCB RaPID Assay Sample Extraction Kit (Modern Water) and PCB analysis using Enzyme Linked Immunosorbent Assay (ELISA)—PCB RaPID Assay (Modern Water). PCB extraction and level determination were performed according to the manufacturer's instructions (https://www.modernwater.com/pdf/MW_Factsheet_Rapid-Assay_PCB.pdf).

The PCB RaPID Assay kit applies the principles of ELISA to the determination of PCB and related compounds (EPA SW-846 Method #4020). The test sample is added to a disposable test tube with an enzyme conjugate, followed by paramagnetic particles bound to antibodies specific to PCB. Both the PCB in the sample and the enzyme-labeled PCB (the enzyme conjugate) compete for the antibody binding sites on the magnetic particles. After a short incubation period, the PCB and the labeled PCB analog will
bind to the particles at proportions reflecting their concentrations in the suspension. A magnetic field is then applied to hold the particles in the tube and the unbound reagents are decanted. After decanting, the particles are washed with Washing Solution. The presence of PCB is detected by adding the enzyme substrate (hydrogen peroxide) and the chromogen (3,3′,5,5′-tetramethylbenzidine). The enzyme labeled PCB analog bound to the PCB antibody catalyses the conversion of the substrate/chromogen mixture to a colored product. After an incubation period, the reaction is stopped and stabilized by the addition of acid. Since the labeled PCB (conjugate) was in competition with the unlabeled PCB (sample) for the antibody sites, the color developed is inversely proportional to the concentration of PCB in the sample. The absorbance was measured at 450 nm using a SDI Differential Spectrophotometer.

A fuller description of sample preparation was also given previously by *Wyrwicka, Steffani & Urbaniak (2014)*.

## Plant material

Cucumber seeds (*Cucumis sativus* L.) cv. Cezar were germinated in Petri dishes for seven days and the seedlings were planted in the control or amended soil. The samples were grown in a growth chamber at of $23 \pm 0.5$ °C under a 16 h light/8 h dark cycle, 250 $\mu$mol m$^{-2}$ s$^{-1}$ photon flux density, during the light period, and 60% relative humidity. Five-week old plants with five fully-expanded leaves were used for subsequent analysis. All biochemical analyses were carried out on the physiologically second, third and fourth leaves (from the bottom) of the control and treated plants. The leaves were harvested in the middle of the 16-hour light period.

## Preparation of enzyme extracts from leaf tissues

The leaves of the cucumber plants were ground (1:10, w/v) in an ice-cold mortar using 50 mM sodium phosphate buffer (pH 7.0) containing 0.5 M NaCl, 1 mM EDTA, and 1 mM sodium ascorbate. The slurry was filtered through two layers of Miracloth. The filtrates of homogenized cucumber leaves were then centrifuged (15,000 g × 15 min). After centrifugation, the supernatant was collected and APx and CAT activities as well as protein content and degree of lipid peroxidation were measured.

## Enzyme assay

APx activity [EC 1.11.1.11] was assayed following the oxidation of ascorbate to dehydroascorbate at 265 nm ($\varepsilon = 13.7$ mM$^{-1}$ cm$^{-1}$) according to *Nakano & Asada (1981)* with some modifications, using a Unicam UV 300 UV-Visible spectrometer (Unicam Limited, Cambridge United Kingdom). The assay mixture contained 50 mM sodium phosphate buffer pH = 7.0, 0.25 mM sodium ascorbate, 25 $\mu$M H$_2$O$_2$ and the enzyme extract (5–10 $\mu$g protein). The addition of H$_2$O$_2$ started the reaction. The obtained values were compared with those of another reaction mixture without the enzyme extract to correct for non-enzymatic oxidation of ascorbate. The enzyme activity was expressed in $\mu$mol ascorbate mg$^{-1}$ protein.

CAT activity [EC 1.11.1.6] was measured spectrophotometrically according to *Dhindsa, Plumb-Dhindsa & Thorpe (1981)*. A reaction mixture composed of 50 mM sodium

phosphate buffer (pH = 7.0), 15 mM $H_2O_2$ and the enzyme extract (5–10 µg protein) was used. The decomposition of $H_2O_2$ ($\varepsilon = 45.2$ mM$^{-1}$ cm$^{-1}$) was measured at 240 nm using Unicam UV 300 UV-Visible spectrometer (Unicam Limited, Cambridge United Kingdom). CAT activity was expressed in mmol $H_2O_2$ mg$^{-1}$ protein.

## Protein content

The protein content was determined according to *Bradford (1976)* with standard curves prepared using bovine serum albumin. Samples were assayed using a spectrophotometer (Helios Gamma, Thermo Spectronic, Cambridge, UK). The protein content was given as mg g$^{-1}$ fresh mass of the original plant tissue.

## Degree of lipid peroxidation (TBARS)

Concentration of lipid peroxides was estimated spectrofluorometrically (F-2500 Fluorescence Spectrophotometer; Hitachi, Limited, Tokyo Japan) according to *Yagi (1982)*, by measuring the content of 2-thiobarbituric acid reactive substances (TBARS). The concentration of lipid peroxides was calculated in terms of 1,1,3,3-tetraethoxypropane, which was used as a standard and expressed in nmol g$^{-1}$ fresh mass.

## Determination of chlorophyll content

Whole leaves were homogenized (1:5 w/v) in an ice-cold mortar using 50 mM sodium phosphate buffer, pH 7.0, containing 0.5 M NaCl, 1 mM EDTA and 1 mM sodium ascorbate. Crude homogenate obtained after filtration was assayed for chlorophyll content according to *Porra, Thompson & Kriedmann (1989)*. The absorbance of clear supernatant obtained after centrifugation (5,000 g × 5 min) of 80% acetone extract was measured at 663 and 645 nm using a Helios Gamma spectrophotometer (Thermo Spectronic, Cambridge, UK). The concentration of chlorophyll was expressed as mg g$^{-1}$ fresh mass.

## Statistical analysis

Linear regression analysis was used to describe the relationships between PCB concentration in soil and amendments dose separately for SS and SED. To analyze whether both regression lines differed significantly, differences in slopes (b coefficient) was tested using a $t$-test and if the null hypothesis were not rejected, the differences in intercepts (a coefficient) were also tested ($t$-test) (*Zar, 2010*). The effect of cucumber plant cultivation on the PCB content in soil amended with SS and SED was analyzed by two-way ANOVA (repeated measures) with type of soil amendments and their dose as factors.

Six to eight individuals were collected from the cultivated plants and used for the experiment, hence the replicate number of plant biochemical parameters varies from six to eight ($n = 6 - 8$). Two-way analysis of variance (ANOVA II) was used to test the effects of both SS and SED and their content in the soil on plant biochemical parameters (i.e., protein concentration, APx, CAT, TBARS and chlorophyll concentration as well as chlorophyll a/b ratio). Before this, compliance with ANOVAs assumptions i.e., the normal distribution and the homoscedicity of variance had previously been checked by the Shapiro–Wilk test and Levene's test, respectively. If the analysis of variance showed a significant effect of any of the factors, a multiple-comparison least significant difference (LSD Fisher) *post hoc* test

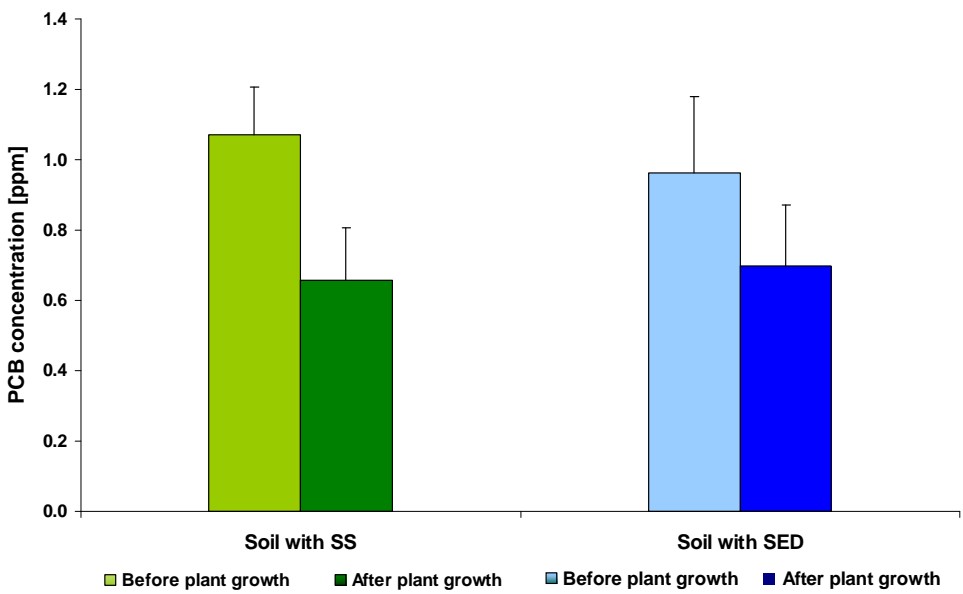

**Figure 1** The average PCB concentration (average ± SD) in the soil fertilized with SS from LM WWTP (green bars) and SED from SSBS (blue bars), before and after five weeks of cultivation.

was used (*Zar, 2010*). All analyses were performed using STATISTICA 13 software (*Dell Inc, 2016*).

# RESULTS

In general, the PCB concentration (average ± SD) in soil amended with SS ($0.931 \pm 0.284$) and in the soil enriched using SED ($0.898 \pm 0.245$) did not differ significantly ($t = 0.6226$; $df = 8$; $p = 0.5436$). In both types of soil amendments, the PCB concentration was related to SS/SED dose; the equations describing these linear relationships were as follows: PCB $= 0.7043 + 0.0505 \times$ amendments dose ($r^2 = 0.6139$; $p = 0.0214$) for SS; PCB $= 0.6206 + 0.0507 \times$ amendments dose ($r^2 = 0.8361$; $p = 0.0015$) for SED. Covariance analysis revealed a lack of significant differences in b (slope) coefficients ($t = 0.01483$; $df = 12$; $p = 0.9883$) and a (intercept) coefficients ($t = 1.1113$; $df = 13$; $p = 0.2865$). After five weeks of cultivation, the mean reduction of PCB was 38.47% for the SS-treated soil and 27.62% for the SED-treated soil (Fig. 1). However, two-way analysis of variance (repeated measures ANOVA II) showed that the reduction of PCB content in soil was significant ($F = 22.1443$; $df = 1, 8$; $p = 0.0015$) and did not depend on the type of amendments used ($F = 0.2091$; $df = 1, 8$; $p = 0.6596$) but confirmed the strong effect of the SS/SED dose ($F = 10.1274$; $df = 3, 8$; $p = 0.0042$) (Figs. 2A and 2B). Moreover, there were no types of significant interactions between analyzed factors (Table 2). *Post hoc* comparisons (the LSD test) revealed significant reductions of PCB concentration for all doses of SS (Fig. 2A) but no significant effect of remediation was observed in the case of SED (Fig. 2B).

Changes in soluble protein content in cucumber leaf tissues were dependent on both the type and the dose of the applied amendment (Fig. 3). Two-way analysis of variance
**A**

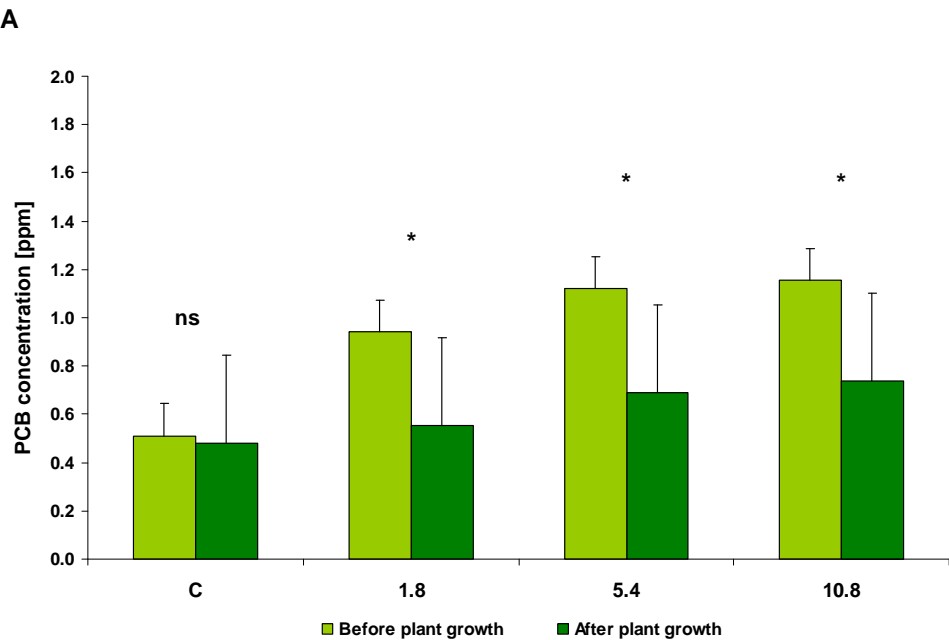

**B**

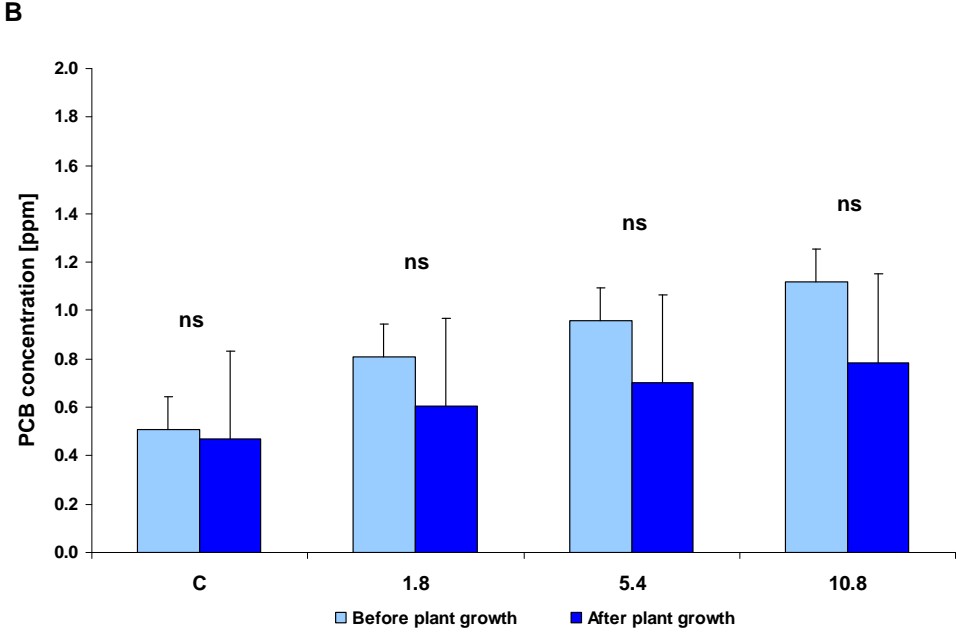

**Figure 2  The PCBs concentration (average ± 95% CL) in the soil fertilized with SS from LM WWTP (green bars, Fig. 2A) and SED from SSBS (blue bars, Fig. 2B), before and after five weeks of cultivation.** Asterisks denote a significant difference at the 0.05 level and 'ns' when $p > 0.05$ (the LSD Fisher *post hoc* test).

**Table 2  Two-way analysis of variance (repeated measures ANOVA II) results to test the effects of the dose of applied amendments (0, 1.8, 5.4 and 10.8), the type of soil amendments (SS and SED) on decreases in soil PCB content after five weeks of cultivation.**

| Effect | df | MS | F | p |
|---|---|---|---|---|
| Dose of applied amendments (A) | 3 | 0.3203 | 10.1294 | 0.0042 |
| Soil amendment type (B) | 1 | 0.0066 | 0.2091 | 0.6596 |
| A × B | 3 | 0.0027 | 0.0862 | 0.9656 |
| Error | 8 | 0.0316 | | |
| PCB decrease (C) | 1 | 0.5544 | 22.1443 | 0.0015 |
| C × A | 3 | 0.0481 | 1.9198 | 0.2049 |
| C × B | 1 | 0.0231 | 0.9232 | 0.3648 |
| C × B × A | 3 | 0.0040 | 0.1580 | 0.9216 |
| Error | 8 | 0.0250 | | |

(ANOVA II) showed the effect of the type of amendments used on protein content ($F = 15.736$; $df = 1, 46$; $p = 0.000253$) as well as the effect of SS/SED dose in the soil on soluble protein content ($F = 9.651$; $df = 3, 46$; $p = 0.000047$). In the case of SS, the multiple comparisons (LSD Fisher) test revealed significant effect of its dose on the leaf protein content. An increasing trend was observed regarding the soluble protein content in leaf tissues together with used dose: 116% of control value for 1.8, 129% for 5.4 and 136% for 10.8. However, the use of SED did not significantly alter the content of soluble protein in cucumber tissues: the only significant difference was found between the 1.8 and 10.8 SED doses. Moreover, a significant interaction (type of amendment × SS/SED dose) was also noted ($F = 4.340$; $df = 3, 46$; $p = 0.00894$). The multiple comparisons (LSD Fisher) test showed that this interaction was significant: no differences in leaf protein content were observed between control, low (1.8) and medium (5.4) of SED doses.

Two-way analysis of variance (ANOVA II) did not show the influence of the type of amendments used on the total chlorophyll content value ($F = 1.342$; $df = 1, 46$; $p = 0.2527$), nor did it indicate that SS/SED dose in the soil had any effect on the value of total chlorophyll content ($F = 0.531$; $df = 3, 46$; $p = 0.664$) (Table 3).

In contrast to total chlorophyll content, the chlorophyll a/b ratio depended significantly on the type of amendments used ($F = 12.05$; $df = 1, 40$; $p = 0.001257$) and the dose of amendments added to the soil ($F = 7.77$; $df = 3, 40$; $p = 0.000334$). The multiple comparisons (LSD Fisher) test showed that the highest dose of both SS and SED used increases the chlorophyll a/b ratio up to 105.7% and to 103.1% of the control value (Table 3). An analysis of both total chlorophyll content and chlorophyll a/b ratio revealed no statistically significant interactions with the type of amendments used and its dose in the soil.

Neither SS nor SED enrichment caused any significant change in the TBARS content, which represents an indicator of oxidative lipid damage (Fig. 4). Two-way analysis of variance (ANOVA II) did not show the influence of the type of amendments used on TBARS value ($F = 0.4392$; $df = 1, 40$; $p = 0.511$). The analysis also did not confirm that SS/SED dose in the soil influenced TBARS level ($F = 2.370$; $df = 3, 40$; $p = 0.085$). The

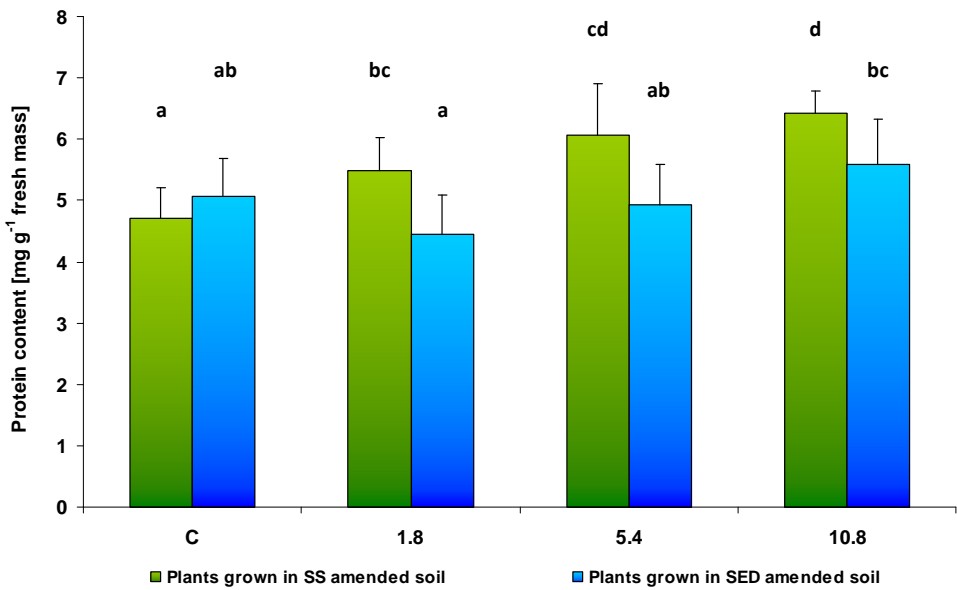

**Figure 3** **Soluble protein content in control cucumber leaves and in leaves of plants grown on soil amended with SS from LM WWTP (green bars) or SED from SSBS (blue bars) at different doses after five weeks of cultivation.** The same letters denote groups that did not statistically differ (the LSD Fisher *post hoc* test).

**Table 3** **The effect of the application of SS from LM WWTP and SED from SSBS on total chlorophyll content, as well as chlorophyll a/b ratio (both average ± SD) of cucumber leaf tissues after five weeks of cultivation.**

|  | Dose of applied amendments | Soil with | |
| --- | --- | --- | --- |
|  |  | SS | SED |
| Total chlorophyll [mg g⁻¹ f.m.] | C | $2.380^{ab} \pm 0.233$ | $2.467^{ab} \pm 0.481$ |
|  | 1.8 | $2.495^{ab} \pm 0.240$ | $2.485^{ab} \pm 0.096$ |
|  | 5.4 | $2.624^{ab} \pm 0.447$ | $2.315^{a} \pm 0.123$ |
|  | 10.8 | $2.656^{b} \pm 0.242$ | $2.491^{ab} \pm 0.397$ |
| Chlorophyll a/b ratio | C | $2.083^{a} \pm 0.065$ | $2.023^{b} \pm 0.019$ |
|  | 1.8 | $2.082^{a} \pm 0.024$ | $2.067^{ab} \pm 0.020$ |
|  | 5.4 | $2.075^{ab} \pm 0.036$ | $2.062^{ab} \pm 0.036$ |
|  | 10.8 | $2.201^{c} \pm 0.093$ | $2.087^{a} \pm 0.055$ |

**Notes.**
The same letters denote groups that did not statistically differ (the LSD Fisher *post hoc* test).

multiple comparisons (LSD Fisher) test revealed SS only demonstrated a significant effect at the 5.4 dose, where TBARS value achieved 130% of control. There were no statistically significant interactions between the analyzed factors.

Two-way analysis of variance (ANOVA II) did not show the influence of the type of amendments used on APx activity, a key antioxidant enzyme in plants ($F = 0.2915$; $df = 1$, 40; $p = 0.5923$). Contrary, dose-dependent changes in the activity of APx were observed after both SS and SED application ($F = 13.1692$; $df = 3$, 40; $p = 0.000004$) (Fig. 5). The

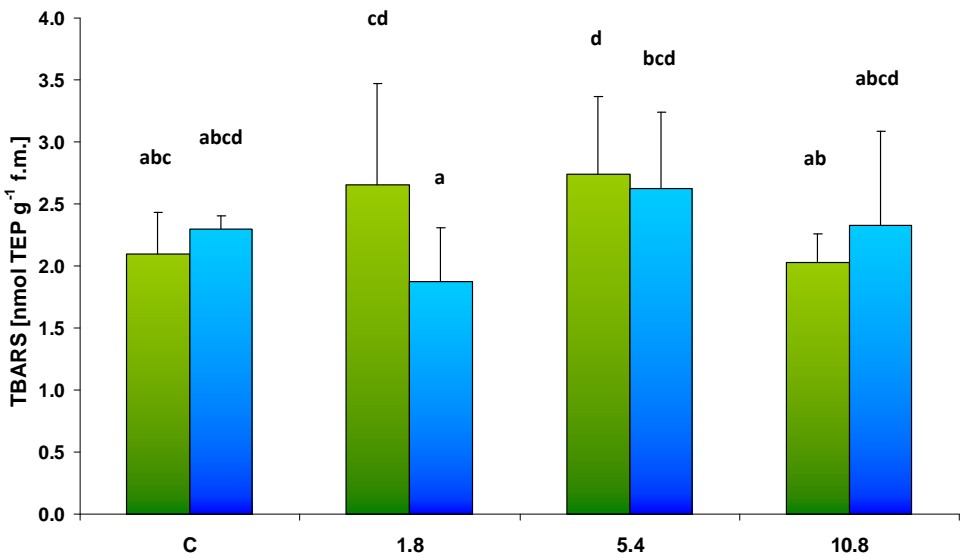

**Figure 4** **TBARS content in control cucumber leaves and in leaves of plants grown on soil amended with SS from LM WWTP (green bars) or SED from SSBS (blue bars) used in different doses after five weeks of cultivation.** The same letters denote groups that did not statistically differ (the LSD Fisher *post hoc* test).

multiple comparisons (LSD Fisher) test revealed that SS dose had a significant effect on enzyme activity: APx activity was strongly reduced to 48%, 49% and 45% of control value in the low (1.8), medium (5.4) and high (10.8) doses, respectively. Similar changes were observed after applying SED, where a decrease in APx activity was observed for the medium (5.4) and high (10.8) doses, these being 66% and 55% of control values, respectively. There were no statistically significant interactions between the analyzed factors.

The activity of CAT, the other enzyme involved in antioxidant defense, depended on the type of soil amendment ($F = 119.198$; $df = 1, 40$; $p = 0.000000$) and SS/SED dose ($F = 7.834$; $df = 3, 40$; $p = 0.000314$). Two-way analysis of variance also showed significant interaction between factors ($F = 6.003$; $df = 4, 40$; $p = 0.00178$). However, the multiple comparisons (LSD Fisher) test revealed that SS dose had a significant effect on CAT activity, but no difference was observed between control and the lowest dose (1.8) and between medium (5.4) and high (10.8) doses; in addition, SED dose did not significantly affect enzyme activity (Fig. 6).

## DISCUSSION

The most important task of phytoremediation is to reduce the levels of toxic substances in the water, sediment, soil or air in a particular environment. Many plants can be used for phytoremediation; however, their physiological condition will be affected to varying degrees by the toxicity of the environment, and this in turn will influence their removal efficiency. With this in mind, the members of the Cucurbitaceae are particularly good

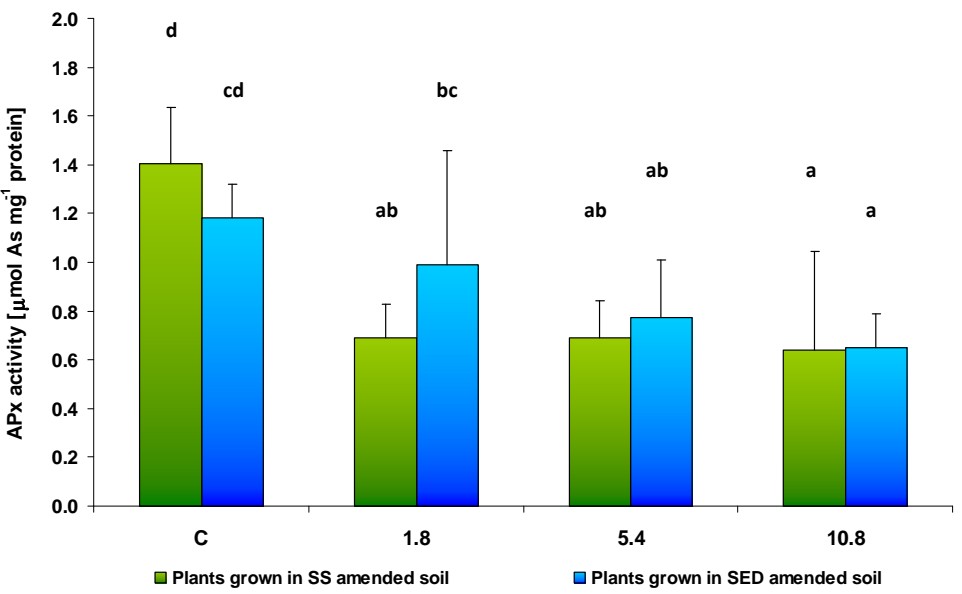

**Figure 5** **APx activity in control cucumber leaves and in leaves of plants grown on soil amended with SS from LM WWTP (green bars) or SED from SSBS (blue bars) used at various doses after five weeks of cultivation.** The same letters denote groups that did not statistically differ (the LSD Fisher *post hoc* test).

candidates for the phytoremediation of PCBs from the substrate (*Mattina, Iannucci-Berger & Dykas, 2000*; *White, 2002*).

Despite being banned in the 1970s because of their negative impact on the environment and human health, PCBs are highly persistent and require removal from the environment. Of the various PCB reclamation methods available, such as incineration and landfilling, a cost-effective and environmentally-friendly approach is phytoextraction (*Low et al., 2010*). The ability of plants to remediate PCBs from the soil is influenced by the properties of the mixture of PCB congeners contained in the substrate and the possibility of their uptake by plants. It has been shown that the degree of chlorination, log $K_{ow}$ and molecular size are the most important factors determining the possibility of absorption and transport for PCBs and other hydrophobic POPs (*Greenwood, Rutter & Zeeb, 2011*).

Our present findings indicate that irrespective of whether SS or SED was added to the soil, the PCB concentration in the amended soil at the beginning of experiment was similar and dependent on the dose of applied amendments. In addition, after five weeks of cucumber cultivation, lower PCB levels were observed in all soil samples; however, this effect was greater in the SS amended soil, where as much as a 41% decrease in PCB concentration was found compared to the initial value. These values suggest a stronger reduction in PCB content than noted in previous studies. For example, *Qin, Brookes & Xu (2014)* report that 60-day cultivation of cucumber plants reduced the PCB content by 19.5% compared to unplanted soil. The authors suggest that one of the important factors determining the effectiveness of phytoremediation may be the presence and composition of microbial communities responsible for PCB degradation. The rhizosphere is a space in

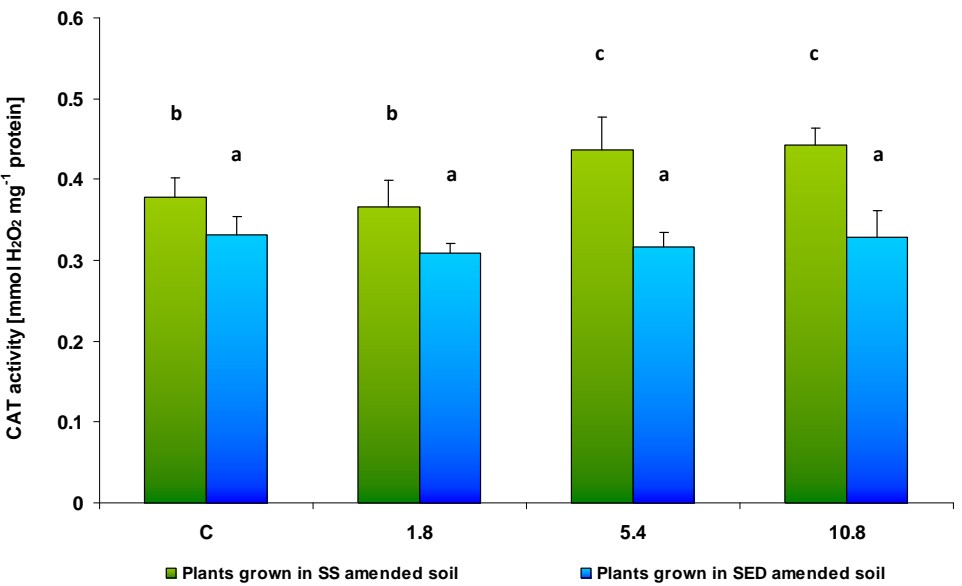

**Figure 6** **CAT activity in control cucumber leaves and in leaves of plants grown on soil amended with SS from LM WWTP (green bars) or SED from SSBS (blue bars) used in different doses after five weeks of cultivation.** The same letters denote groups that did not statistically differ (the LSD Fisher *post hoc* test).

which root exudates are released to the soil, and these may selectively foster the growth of PCB-degrading bacteria. The effectiveness of PCB remediation may, therefore, not only be conditioned by the species, or variety of the plant involved in phytoremediation, but also by the composition and origin of the sediments containing PCBs. As the SS and the SED used in the present study most likely possess different microbial compositions, it is not surprising that they were associated with different degrees of PCB reduction.

Differences between the type of applied amendment (SS or SED), influencing the degree of decrease of PCB concentration in the studied soil (revealed by the LSD test), may indicate that the ability to remediate PCBs from amended soil by cucumber plants may depend on the origin of the amendments used, and hence on the composition of the soil. It can not be ruled out that the ability to remediate PCBs may depend on the presence of other toxic substances in the substrate that can cause adverse effects on plants. A study of the remediation potential of zucchini (*Cucurbita pepo* L.) in soil samples treated with PCDD/PCDF-contaminated SS or SED found the PCDD/PCDF content to be up to 2.7 times greater in the soil treated with SS than in that treated with SED (*Urbaniak et al., 2016*). However, the presence of these compounds in the SS-treated soil did not reduce its level of phytoremediation; in fact, the level was even greater than in the SED soil.

While the cucumber plants treated with SS demonstrated gradual increases in total soluble protein content with increasing doses of the applied sludge, no significant changes were found in plants growing on soil with SED. An elevated total protein content may indicate that the presence of SS has a beneficial effect on the physiological condition of the

plant. From the physiological point of view, it is known that the soluble protein content in plants decreases during the course of senescence (*Camp et al., 1982*; *Romanova et al., 2016*). This is related to the initiation of the N-remobilization process, during which proteinase activity increases, leading to the degradation of proteins to peptides or amino acids, which are then transported to the growing organs in the phloem sap (*Girondé et al., 2015*). Hence, the increase in soluble protein content observed in cucumber tissues growing in SS amended soil may indicate a delay of the processes of senescence. Protein content in plant tissues may also increase due to abiotic stresses such as heat or drought (*Grigorova et al., 2011*). In the case of cucumber plants studied in this work, the thesis about increasing the protein content resulting from stress seems unlikely, if only due to the lack of a significant increase in TBARS content in plants treated with SS and SED, which is a measure of oxidative damage of lipids as well as due to the lack of lowering total chlorophyll content in leaf tissues.

This increase in total soluble protein content in the leaf tissues of cucumber cultivated on soil treated with SS, combined with the apparent lack of influence of SS and SED contamination on total chlorophyll content, i.e., no decrease was observed, may be another indicator that the examined plants are in good physiological condition and the applied amendments may have a strong fertilizing effects. The results of both the total protein and chlorophyll content obtained in this study show that senescence progresses faster in control plants than in plants fertilized with SS as well as the use of SED also do not intensify the processes associated with senescence.

The physiological condition of the plant can also be evaluated my measuring chlorophyll a/b ratio. It was found that chlorophyll a/b ratio increased slightly in the plants treated with the highest dose (10.8) of SS, as well as with SED, in comparison with control. As mentioned earlier, total chlorophyll content together with chlorophyll a/b ratio may be indicators of the initiation of senescence process in plants and gradual changes in the biochemical processes associated with them. A decrease in leaf chlorophyll content is one of the first symptoms of plant senescence, which is characterized by the breakdown of thylakoid membranes and the degradation of thylakoid-bound proteins. The presence of the aging process is further confirmed by a depressed chlorophyll a/b ratio. For example, in a study of *Arabidopsis thaliana* plants, the chlorophyll a/b ratio was found to decrease linearly with natural leaf senescence, as did total chlorophyll content; differential degradation of chlorophyll a and chlorophyll b was also observed during senescence, resulting in changes in chlorophyll a/b ratio (*Nath et al., 2013*). In the case of the cucumber plants studied in the present work, it can be assumed that the addition of SS and SED not only acts as a fertilizer, but also serves a protective function against the accelerated aging process. Our findings also highlight another important point: cucumber plants may well be resistant to toxic effects of substances contained in SS.

TBARS content is a commonly-used indicator of oxidative stress following environmental stress (*Srivastava, Sinha & Sharma, 2017*). In the present study, both after applying SS and SED, there were no changes in TBARS content in cucumber leaves that could indicate oxidative damage of lipids. The only exception is the increase in the value of this parameter (up to 130% of control) after application of SS at the medium (5.4) dose.

*Page et al. (2014)* report lower application rates of compost-like-output result in nitrate leaching, and note that this could be a limiting factor in the use of sludge as a nutrient source. It is also possible that the advantages of sludge application associated with nutrient content are outweighed by the presence of toxic compounds when applied at low doses: Higher doses of sludge are richer in nutrients, and this compensates for the effects of the presence of harmful substances. Similarly, SS was found to have a protective effect on alfalfa plants (*Medicago sativa* L. cv. Aragon) subjected to additional drought stress: Its addition led to a significant reduction malondialdehyde in nodules resulting from lipid peroxidation, while this number was strongly increased in the nodules of untreated plants subjected to drought (*Antolín, Muro & Sánchez-Díaz, 2010*). A similar beneficial result was obtained by *Tartoura & Youssef (2011)* regarding the use of compost for the cultivation of squash (*Cucurbita pepo* L. cv. Eskandarany) exposed to low temperature conditions, with both oxidative damage of lipids and ROS generation being markedly reduced following compost application.

Our present findings also indicate differences in the plant response to the use of SS and SED. Although the soil PCB concentrations at the beginning of the experiment did not differ significantly and depended only on the dose of the amendments used, the response of plants growing on SS and SED-amended soil varies. It is very likely that such a situation would be attributed to the composition of soil after addition of SS or SED. It can not be ruled out that SS is more toxic than SED and the two can vary to a great degree regarding their remaining content. Earlier studies carried out on willow plants indicate that the composition of SS and its impact on plants is dependent on the size of the sewage treatment plant (*Wyrwicka & Urbaniak, 2018*). For example, unlike the sludge produced by large, industrialized urban centers, like that used in the present work, sludge from plants operating in small agglomerations does not typically contain large amounts of substances harmful to plants, such as heavy metals.

APx is an enzyme characteristic of plant tissues. It is characterized by high substrate specificity and is sensitive to substrate concentration; during its absence in the surrounding solution, the enzyme is inhibited. This enzyme may also be inactivated by $H_2O_2$. It is also an important element of the ascorbate-glutathione cycle, which maintains the reduction-oxidation balance in plant cells (*Asada, 1992*). Of the parameters measured in the present study, APx was found to display the greatest changes in activity, with decreases being observed both after applying SS and SED. In the case of SS, the lowest applied dose (1.8) already inhibited the enzyme activity (up to 48% of control), whereas after the SED application, the APx activity decreased only when the medium (5.4) and high (10.8) doses were used. This may suggest that both SS and SED may contain compounds that inhibit APx activity and it cannot be ruled out that $H_2O_2$ would accumulate in the cells under these conditions. On the other hand, while APx activity was significantly reduced, an increase in CAT activity was observed in the plant tissues exposed to the same dose (5.4 and 10.8) of SS present in soil. It is possible that the increase in CAT activity has a compensatory effect on inhibited APx activity in investigated cucumber tissues. Furthermore, CAT activity, whose primary function it is to remove $H_2O_2$ from cells, in tobacco cell suspension culture

exposed to PCDD/Fs showed an upward trend with increasing PCDD/Fs concentration (*Zhang et al., 2012*).

The decrease in APx activity observed herein may also be attributed to a lack of ascorbate, its substrate, which may have been consumed elsewhere by non-enzymatic antioxidative reactions. Our observations regarding APx activity and the total protein content can be compared with the those of *Aki, Güneysu & Acar (2009)*, who studied the reaction of four plant species, *viz.* tomato, pepper, bean and broad bean, on industrial wastewater. Total protein content increased in all of the tested plants during the experiment, while peroxidase activity decreased following exposure.

The elevated CAT activity observed in the present study after the application of the medium (5.4) and high (10.8) doses of SS could be result of an efficiently functioning antioxidative system in the cucumber tissues: It can be assumed that increased CAT activity protected the tissues of the investigated cucumber plants from oxidative damage, which can be proved only by a small increase in the TBARS content in one of the applied concentration variants (5.4). As it appears from the results obtained, after using a higher dose (10.8) of SS, elevated CAT activity was probably effective in protecting plant tissues from the effects of oxidative stress, because in any of the variants elevated TBARS content was found. It cannot be ruled out that changes in CAT activity after SS are related to the size of the dose of amendment. A previous study on wheat (*Triticum durum*) exposed to different doses of SS found CAT activity in the leaves to be unaffected when the sludge was applied in small doses, but significantly elevated at higher doses (*Lakhdar et al., 2009*).

The increase in total protein content and chlorophyll a/b ratio observed in the present study, as well as the changes in CAT activity and lack of the increase of TBARS content (found only in one case), indicate that the plants fertilized with SS fared better than the unfertilized controls, and this suggests that cucumber plants are well suited to phytoremediation of PCBs. Our results suggest that cucumber plants are able to trigger a defensive response based on the action of the antioxidant system, such as CAT activity, that allows them to survive adverse environmental conditions and resist the effects of toxicity.

## CONCLUSIONS

Our findings indicate that the cucumber is perfectly suitable for reducing PCB content in soil treated with SS or SED in a relatively short time, while appearing relatively resistant to the toxic substances present in the applied amendment. The data concerning total protein content, total chlorophyll content and chlorophyll a/b ratio indicates that plants treated with SS or SED fared better than untreated controls, with the greatest response displayed by those treated with SS. The cucumber plants treated with SS or the SED possessed an effective, functioning antioxidative system which protected the plants against the occurrence of oxidative damage. Therefore, cucumber plants are ideal candidates for the phytoremediation of this type of soil pollution. However, further studies are needed to integrate the knowledge regarding the mechanism of PCB action and activation of plant protective systems, as well as the interdependence between the plants and microbes responsible for PCB degradation, in order to improve the efficiency of the phytoremediation process and adapt it to practical applications.

### Funding
This work was supported by the Ministry of Science and Higher Education program under the name "Iuventus Plus" for the years 2015–2017 granted on the basis of decision number 0492/IP1/2015/73 (http://www.nauka.gov.pl/) and University of Lodz Grant No. B17 11 000 000 052.01. There was no additional external funding received for this study. The funders had no role in study design, data collection and analysis, decision to publish, or preparation of the manuscript.

### Grant Disclosures
The following grant information was disclosed by the authors:
Ministry of Science and Higher Education program: 0492/IP1/2015/73.
University of Lodz: B17 11 000 000 052.01.

### Competing Interests
The authors declare there are no competing interests.

### Author Contributions
- Anna Wyrwicka conceived and designed the experiments, performed the experiments, analyzed the data, contributed reagents/materials/analysis tools, prepared figures and/or tables, authored or reviewed drafts of the paper, approved the final draft, preparation of the substrate, plant cultivation, linguistic correction.
- Magdalena Urbaniak conceived and designed the experiments, performed the experiments, contributed reagents/materials/analysis tools.
- Mirosław Przybylski analyzed the data.

### Data Availability
The raw data is available as a Supplemental File.

### Supplemental Information
Supplemental information for this article can be found online at http://dx.doi.org/10.7717/peerj.6743#supplemental-information.

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
