# Peer review of "The response of cucumber plants (Cucumis sativus L.) to the application of PCB-contaminated sewage sludge and urban sediment"

_PeerJ, doi:10.7717/peerj.6743_

## Round 0.1 · original submission · Minor Revisions

Please respond to all reviewer comments, but pay particular attention to 2 significant comments. One, it appears that plants grown in amended soil did not show a statistically supported difference from control. Therefore, statements and discussion should be modified accordingly. Two, a description of how PCBs were determined is needed.

Reviewer 1 ·

Basic reporting

The Ms by Wyrwicka and Urbaniak is written in a clear way, easy to read and with appropriate reference to data in figures and tables. Overall, the structure of the work sounds valid. The state of the art is correctly depicted and the scientific context (Phytoremediation) appropriately described. Literature is updated and properly reported and discussed.

Experimental design

The topic of the Ms fits well with the scope of the journal. The research questions are properly defined and addressed. The investigation is sound with a good technical approach and the methods appropriately described and applied.

Validity of the findings

The novelty of the work and how it can contribute to the advance of the knowledge on the topic are sufficiently discussed. The data set is properly focused to sustain the essential hypothesis on which the work is based. The statistical approach needs a revision following the comments reported in the general comments section. Conclusion should be more consistent with the statistically correct data interpretation in order avoiding unidentified speculation.

Additional comments

As reported in the previous sections, the work by Wyrwicka and Urbaniak is certainly of interest as highlighting an interesting approach and bringing useful information on the topic of the phytoremediation of PCB present in the sludge or urban sediments.
My main concern is referred to the statistical treatment of data that in my opinion should be accurately revised. Namely, following the statistical approach performed by authors (see specific comments below reported), it seems that most of the data referred to plants grown in amended soil did not show a statistically supported difference from control. Therefore, a lot of statements made by authors to present and discuss data are not accurate and should be modified accordingly. Specific comments are reported below. For your convenience, I numbered the page starting from the abstract one.
Abstract
P1 L12 Not clear if the treatment is only referred to sludge or also the sediments or both together. Please clarify.
P2 L1 This statement is not clear. Did authors refer to heavy metals possibly present in the sludge? I suggest to reformulate the sentence avoiding to mention other compounds that should be quantified.
Introduction
P3 L9 As the reference “Urbaniak et al., 2017” is at first appearance I suggest to denominate it as “a” rather than “b”.
P4 L8 I wonder whether no food plants could be profitably exploited for such scope instead of plants that can enter the food chain.
P5 L12-13 I think that the aim of the work can be better summarized.
M&M
P5 L24 A brief description of the main soil parameters could be reported.
P5 L27 I suppose that such doses are referred to wet matter while authors used dried material. Did authors take into account this issue?
P6 L12 Add “the apex of” before “the control” if this detail is correct.
P7 L28-32 Not clear about six replicates if n= 6-8. Please correct or specify. About the statistical approach, I wonder whether authors did not use ANOVA followed by a statistical test to separate means. Did data fail to pass normality test?
P8 L1-2 Assuming that statistically different data were displayed by the “p” factor as in Figs. 2, 4 and 5, the statement referred to data on Fig. 1 is not appropriate as no differences among such data are reported. Such aspect sounds a bit strange as different doses of sludge/sediments were applied to soil. Moreover, PCB concentration in control soils seems not corresponding to a background value. I suggest authors to revise this part, possibly adopting a different statistical approach in order to better highlighting the reduction of PCB concentration after plant growth.
P9 L2 Also in this case, no statistical outcome allowing for such statement was produced.
P9 L14 I suggest to change the term “gradual” as referred to a trend that was not highlighted.
Discussion
P10 L7-9 See comments on P8L1-2 and revise this part accordingly.
P10 L12-14 Really, to better describe the microbial contribution on PCB concentration decrease, pots with sludge/sediments but without plants could have been tested.
P11 L7-16 Some statements reported in this part are not properly supported by the statistical outcomes reported in figs. and table (e.g. no greater total chlorophyll content was evidenced). I suggest to revise these sentences.
P11 L33-34 TBARS content was not affected by the exposure to the sludge/sediment added soil. This sentence is not consisting with the statement in P13L15.
P12 L31-34 These statements sound not accurate if referred to the statistical assessment of discussed data.
P13 L2 Contrarily, it seems that CAT activity is increasing in plants treated with sludge.
P13 L17 Again, this sentence should be revised in accordance to the statistical evaluation.
Conclusion
I suggest to reformulate the first part of the section to make it better coherent with the discussed data.
Reference
The reference “Lakhdar et al.” and “Srivastava et al.” are not reported in the list. Check the date of the reference “Zhang et al.” and order by alphabet.

Reviewer 2 ·

Basic reporting

The paper is timely, dealing with a worldwide problem of environmental pollution and the usability of sewage sludge and sediments. The text is clear and understandable. There are references covering the field, the introduction is explaining all the aspects of the performed research. While the use of sediments and sludges is not new, the novelty is in evaluation of benefits of activity as fertilizers together with some antistress effect on Cucurbita plants
thus supporting their phytoremediation potential.
The structure of the paper is OK, as well as figures and the table, the raw data are available. There are very few typing errors, see comments for authors, where are suggestions for minor corrections.

Experimental design

The paper presents original primary research which is in agreement with the scope of the journal. It is clearly stated what is the research question, it is meaningful and helps ti identify the dangers and benefits of the use of sludge and sediments
The methods and materials are appropriate and were used correctly. The experimental setup is well described and clear.
What I am missing is more description of how the PCBs were analysed - at least a rough description, the citation of an other paper is OK, but I might appreciate at least a basic info on analytical method also in the text, on used standards, which PCBs were analysed, a sum, indicator congeners only or individual peaks evaluated? It would be enough to add a single paragraph with proper citation.

Validity of the findings

The paper brings new information concerning the application of sewage sludge or sediments, describing the danger of introducing toxicity of soil, but evaluating the advantageous effect of supporting phytoremediation potential by introducing nutrients into the soil. Authors discuss also the delay of senescence of cucumber plants caused by sludge addition. The conclusions and results are matching the experiments, which are statistically confirmed.

Additional comments

The paper brings new information important for evaluation of potential phytoremediation technology. I have only a few comments. As already stated above, one most important question concerns missing more description of how the PCBs were analysed - at least a rough description.
Other minor problem:
on page 7, line 18, probably you meant Miracloth, and not Micracloth.
on page 10, line 14 ... were observed ...
on page 10, line 18-22, you write ... both sludge and sediment elicited CAT... but on line 21it is unclear, no such relationship...
on page 12, line 3 ... soluble protein decreases during senescence... And how is it with effect of stress, e.g. caused by toxicity of compounds from sludge?
FIG 1. ...... explain better, e.g. PCB added within the sediment .., or ...contained after treatment by...
FIG 2. the capture might be more improved, what does it tell.

---

## Round 0.2 · accepted · Accept

Thank you for your efforts to improve your manuscript based on reviewer comments.

# Reviewer 1 ·

Basic reporting

The revised version of the Ms by Wyrwicka et al. was notably improved by properly addressing my remarks, especially those concerning the statistical data assessment. In accordance with the new approach in the statistical data elaboration, results presentation and discussion were satisfactorily modified making the Ms a sound work highlighting the multifaceted responses of cucumber plants to the application of sewage sludge and urban sediments. Therefore, the Ms meets the requirement for publication.

Experimental design

The revised version of the Ms by Wyrwicka et al. was notably improved by properly addressing my remarks, especially those concerning the statistical data assessment. In accordance with the new approach in the statistical data elaboration, results presentation and discussion were satisfactorily modified making the Ms a sound work highlighting the multifaceted responses of cucumber plants to the application of sewage sludge and urban sediments. Therefore, the Ms meets the requirement for publication.

Validity of the findings

The revised version of the Ms by Wyrwicka et al. was notably improved by properly addressing my remarks, especially those concerning the statistical data assessment. In accordance with the new approach in the statistical data elaboration, results presentation and discussion were satisfactorily modified making the Ms a sound work highlighting the multifaceted responses of cucumber plants to the application of sewage sludge and urban sediments. Therefore, the Ms meets the requirement for publication.

Additional comments

The revised version of the Ms by Wyrwicka et al. was notably improved by properly addressing my remarks, especially those concerning the statistical data assessment. In accordance with the new approach in the statistical data elaboration, results presentation and discussion were satisfactorily modified making the Ms a sound work highlighting the multifaceted responses of cucumber plants to the application of sewage sludge and urban sediments. Therefore, the Ms meets the requirement for publication.